# Leaf Mesophyll Mitochondrial Polarization Assessment in *Arabidopsis thaliana*

**DOI:** 10.3390/mps4040084

**Published:** 2021-11-17

**Authors:** Cesar Flores-Herrera, Emilia R. Gutiérrez-Mireles, Manuel Gutiérrez-Aguilar

**Affiliations:** Departamento de Bioquímica, Facultad de Química, Universidad Nacional Autónoma de México, Ciudad Universitaria, México City 04510, Mexico; cesar452.2jg@gmail.com (C.F.-H.); emi.gutierrezm@gmail.com (E.R.G.-M.)

**Keywords:** plant mitochondria, abiotic stress, leaf mesophyll

## Abstract

Plant leaves present an intricate array of layers providing a robust barrier against pathogens and abiotic stressors. However, these layers may also constitute an obstacle for the assessment of intracellular processes, especially when using fluorescence microscopy approaches. Current methods for leaf mitochondrial membrane potential determinations have been traditionally performed in thin mesophyll sections, in isolated protoplasts or in fluorescent protein-expressing transgenic plants. This may limit the amount of information obtained about overall mitochondrial morphology in intact leaves. Here, we detail a fast and straightforward protocol to assess changes in leaf mitochondrial membrane potential associated with mitochondrial dysfunction in the model plant *Arabidopsis thaliana.* This protocol also permits mitochondrial shape, dynamics and polarity assessment in leaves subjected to diverse stress conditions.

## 1. Introduction

Plant mitochondria constitute an important hub for metabolism during development and under stress conditions. For example, plant mitochondrial respiration is of utmost relevance for flower development and pollen tube growth [1,2]. These organelles are also involved in thermogenesis following low-temperature growth conditions [3], while heat stress is a known effector of mitochondrial dysfunction and cell death. For example, Vacca and collaborators have detailed how heat shock can induce mitochondrial cytochrome *c* release in a process attenuated by reactive oxygen species scavenging enzymes [4,5,6]. In these studies, heat shock triggered mitochondrial dysfunction, as assessed by a drop in inner membrane potential (ΔΨ) [6]. Mitochondrial respiratory chain impairments were also evident, as assessed by the loss of succinate oxidizing activity in an ADP-sensitive manner. These studies were performed in tobacco Bright-Yellow 2 (BY2) cells in suspension, which facilitates the assessment of many biochemical parameters including mitochondrial (dys)function. Nevertheless, assessing such parameters in plant tissues such as roots, stems or leaves has proven a much more challenging task. Technical advances for leaf mitochondria imaging include fluorescence microscopy assessment of thin sections or whole plant structures expressing fluorescent proteins targeted to the mitochondria [7]. These technological advances have allowed scientists to understand how plant mitochondria interact with other organelles and how plants can react to deleterious stressors. For example, plant mesophyll mitochondria undergo Mitochondrial Morphology Transition (MMT) upon treatment with oxidizing agents or heat stress. When induced, MMT wreaks havoc and is thought to be a further initiator of pronounced tissue necrosis [8]. Another commonly used approach to assess mitochondrial status is labeling isolated protoplast samples with mitochondria-specific fluorescent dyes. This technique has allowed the assessment of mitochondrial membrane potential perturbations following UV light- or heat-shock-induced damage [9]. However, protoplast isolation can be time consuming and may not reflect accurate pathophysiological scenarios in plants, considering that cell walls are digested [10].

Here, we present a straightforward method to assess leaf mesophyll mitochondrial polarization in situ under normal and stress conditions in the model plant *Arabidopsis thaliana.* This method can be applied to determine mitochondrial status and morphology in plants undergoing the first stages of growth and under conditions where other mitochondrial assessment methods are difficult to implement.

## 2. Experimental Design

### 2.1. Materials


Carbonyl cyanide 4-(trifluoromethoxy)phenylhydrazone (FCCP) (Sigma-Aldrich. St. Louis, MO, USA; Cat. no.: C2920).Silicon Oil (Sigma-Aldrich. St. Louis, MO, USA; Cat. no.: 378437).Perfluorodecalin (PFD), (Sigma-Aldrich. St. Louis, MO, USA; Cat. no.: P9900).Murashige and Skoog (MS) Basal Medium, (Sigma-Aldrich. St. Louis, MO, USA; Cat. no.: M5519).5,5′,6,6′-tetrachloro-1,1′,3,3′-tetraethylbenzimi- dazolylcarbocyanine iodide (JC-1), (Sigma-Aldrich. St. Louis, MO, USA; Cat. no.: T4069).MitoTracker Green FM, (Invitrogen. Waltham, MA, USA; Cat.no.: M7514).Mannitol.Ethylenediaminetetraacetic acid EDTA.K_2_HPO_4_.Tris.NaClO solution.Tween.Scotch^®^ Transparent Tape.Dissecting scissors (Sigma-Aldrich. St. Louis, MO, USA; Cat. no.: Z265969).10 cm Petri plates.Agar.Pipettes.Eppendorf tubes.Microscope slides.Precision tweezers.Coverslips.*Arabidopsis thaliana* Col-0 ecotype seeds (ABRC. Columbus, OH, USA; Cat. no: CS66459).Sunshine Mix 3 (Sungro Horticulture).Miracle-Gro 8 and Miracle Gro Perlite (The Scotts Miracle-Gro Company, Marysville, OH, USA. Model: 70752300).


### 2.2. Equipment


Ambi Hi-Lo incubation chamber (Lab-Line Instruments Inc. Melrose Park, IL, USA, Cat. no.: 3554-35).AmScope T-600C Epifluorescence microscope (AmScope. Irvine, CA, USA; Cat. no.: T600, SKU FK-EPI-NL).Corning LSE low-speed orbital shaker, w/flat platform (Sigma-Aldrich. St. Louis, MO, USA, Cat. No.: CLS6780FP).Autoclave.Laminar flow hood for MS medium preparation.


## 3. Procedure

### 3.1. A. thaliana Growth Conditions (Time to Completion: 3 Weeks)


Steps 2 to 4 should be performed under sterile conditions.Place 10–20 seeds in an Eppendorf tube filled with 1 mL of sterilization solution (3.7% sodium hypochlorite, 0.02% Tween) for 10 min.Wash seeds three times with sterile deionized water.Place seeds in a Petri dish containing sterile ½x MS agar medium [7].Place the Petri dish in an incubator at 22 °C in a 16 h/8 h light/dark photoperiod.Once the seedlings have achieved desired development, transfer to soil or desired substrate (we typically use peat moss Sunshine Mix 3 (Sungro) plus Miracle-Gro-type minerals in a 2:1:1 ratio).
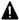
**CRITICAL STEP**. Allow plant growth for no more than 2 to 3 weeks from germination (longer times will result in peeling-resistant leaves).Leaves from plants grown on Petri dishes can be alternatively used for epidermis removal.


### 3.2. Leaf Epidermis Removal (Time to Completion: 3 Weeks)


Select (~0.5 cm) leaves from 3-week-old plants.Adhere a piece of Scotch^®^ Transparent Tape over each leaf adaxial epidermis (Figure 1A).Adhere a piece of Scotch^®^ Transparent Tape to the abaxial epidermis.Remove air bubbles and enhance epidermis adhesion by gently pressing the leaf with one finger.Pull away the Scotch^®^ Transparent Tape slowly. The lower epidermis will be detached.Cut excess tape with scissors to match mesophyll perimeter.All further steps should be performed without any pause.


### 3.3. Leaf Mesophyll Staining (Time to Completion: 45 min)


Place one peeled leaf mesophyll inside an Eppendorf tube containing 1 mL Stain buffer (see Section 5 for buffer preparation instructions).Infiltrate under mild vacuum conditions for 1 min in the absence of light. Make sure the tube cap remains open during the process. Optional: Longer vacuum times can be used to ensure better penetration if needed.Close tube cap and place it in a shaker at 60 rpm for 30 min in the absence of light. Make sure tube contents mix thoroughly.Pipette out or vacuum aspirate Stain buffer without disturbing the leaf (see Section 3.5 for more details).Add 1 mL Wash buffer. (See Section 5 for buffer preparation instructions).Wash the leaf five times with 1 mL Wash buffer.
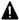
**CRITICAL STEP** Make sure all stain remnants are washed away in the sample and the tube, otherwise keep washing the sample.Remove Wash buffer and add 100 μL PFD above the leaf level.


### 3.4. Leaf Mesophyll Mounting (Time to Completion: 10 min)


Place the leaf with the aid of precision tweezers on a microscope slide and add an even layer of silicon oil around each leaf to form a gasket. Add ~100 μL PFD to fill the gasket.Place a coverslip above, making sure no air bubbles remain inside.Press gently to minimize sample volume.Seal coverslip with transparent nail polish. This is especially relevant when imaging in inverted microscopes.Image samples at 405 nm excitation for enhanced J-aggregate detection at 575–630 nm emission. Use 505–550 nm BP filter for the green channel and 575–630 nm filter for the red channel.


### 3.5. Additional Notes

The overall leaf mesophyll staining workflow is summarized in Figure 1A. Please note that leaves selected for staining should be processed throughout the protocol without delay since these structures tend to deteriorate quickly. Moreover, excess tape removal should be performed matching the mesophyll perimeter so that leaves will not stick to any surface. For quick and efficient buffer removal, we recommend using a vacuum line plugged to a 200 μL pipette tip.

## 4. Expected Results

### 4.1. Leaf Epidermis Removal and Mesophyll Staining

The protocol described herein was developed to detach leaf epidermis without compromising mesophyll optical properties. To achieve this, we first selected and detached leaves from 3-week-old plants for leaf epidermis removal. We used two films of Scotch^®^ Transparent Tape to adhere leaves from both sides and facilitate mesophyll imaging (Figure 1). The remaining mesophylls were incubated with Stain buffer containing JC-1 for 1 min under mild vacuum conditions to ensure complete penetration of the dye in the absence of light and to facilitate imaging using a standard fluorescence microscope (see Reagents Setup section for a note on JC-1 concentration). Samples were then covered with aluminum foil and incubated under mild shaking conditions for 30 min. Excess JC-1 was then removed by washing five times with Wash buffer. Stained mesophylls were mounted in microscope slides coated with silicon oil to form a “gasket”. This gasket was filled with PFD (as detailed in [11]) to enhance optical properties and sealed with a cover slide to avoid leaks (Figure 1). Samples were immediately analyzed under the microscope to determine mitochondrial polarization. Leaf mesophylls not subjected to epidermis removal were largely impermeable to JC-1. In fact, this dye actively accumulated inside stomata in the form of green J monomers (Figure 2A). Following epidermis removal, leaf mesophylls accumulated JC-1 in red particulate J-aggregate patterns corresponding to mitochondria and neighboring chloroplasts. These results show that mitochondria are polarized while closely interacting with chloroplasts (Figure 2B). In fact, mitochondrial wiggling, as described by Oikawa and colleagues, was also evident [7], suggesting the samples retained their characteristic dynamics (not shown). Further treatment with FCCP resulted in leaf mesophyll mitochondria emitting green fluorescence, indicating membrane depolarization due to uncoupling (Figure 2C).

### 4.2. Mitochondrial Polarization Assessment in Heat-Shock-Treated Plants

Heat shock has deleterious effects on plant physiology [12,13]. Upon this abiotic stressor, *Arabidopsis thaliana* displays changes in mitochondrial function consistent with impairments in Complex I, mitochondrial network fragmentation and organellar depolarization [9,14]. In order to test whether the present method can detect heat-shock-induced mitochondrial depolarization, we challenged 3-week-old *A. thaliana* with a 3 h-long heat shock at 37 °C or under optimal temperature conditions at 22 °C (Figure 3). Under control conditions, plants rosettes presented normal morphology and mitochondria from these samples displayed robust red fluorescence, indicating membrane polarization due to red J-aggregate formation (Figure 3A,B). However, when challenged with heat shock conditions, plant rosettes appeared rolled and shrank. Further assessment of mesophyll mitochondria from these samples revealed strong green fluorescence consistent with monomeric JC-1 (Figure 3C,D). Taken together, the results presented herein indicate this method is suitable for detecting mitochondrial polarization status under normal and stress conditions.

## 5. Reagents Setup

Leaf mesophylls are placed in freshly made Stain buffer containing 20 µM JC-1, 330 mM mannitol, 0.1 mM EDTA, 1 mM K_2_HPO_4_ and Tris 10 mM pH 6.8. The 1 mM JC-1 stocks must be kept at −20 °C until used. Wash buffer and Stain buffer are similar in composition except that Wash buffer lacks JC-1. All buffers should be used during the protocol at room temperature. All buffer remains should be discarded. Note: JC-10 (Sigma-Aldrich. St. Louis, MO, USA; Cat. no.: MAK159) can be used instead of JC-1 with similar results, albeit with enhanced dye solubility (not shown).

## Figures and Tables

**Figure 1 mps-04-00084-f001:**
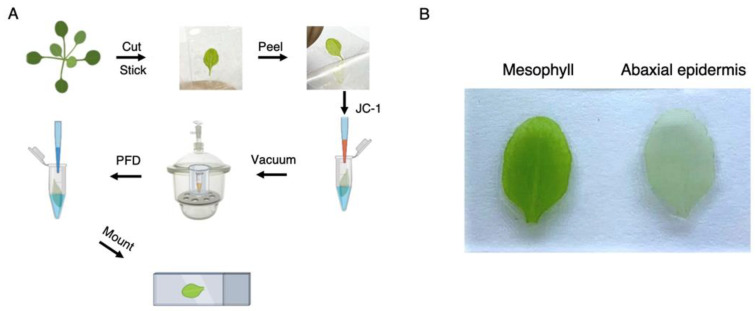
Leaf mesophyll mitochondria potentiometric staining procedure. (**A**) Diagram summarizing sample preparation procedure. (**B**) The abaxial epidermis was successfully detached with Scotch^®^ Transparent Tape for further leaf mesophyll treatment with JC-1.

**Figure 2 mps-04-00084-f002:**
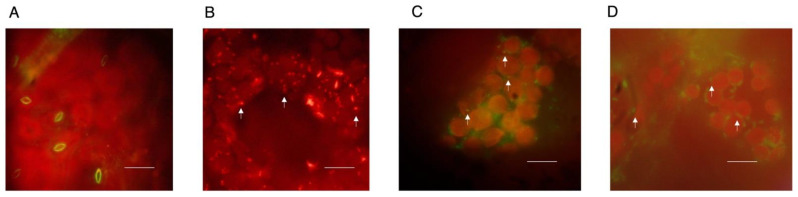
Leaf mesophyll mitochondria are effectively stained with JC-1 upon epidermis removal. (**A**) Leaves were directly stained with JC-1 without epidermis removal and assessed for mitochondrial fluorescence. (**B**) Leaves were treated as detailed in the protocol for leaf mesophyll mitochondria potentiometric staining under control conditions or (**C**) in the presence of 1 µM FCCP. (**D**) Mitochondria were also stained with MitoTracker Green FM as a control for organellar localization. Arrows mark selected mitochondria. Scale bar: 10 µm. Representative images *n* = 5.

**Figure 3 mps-04-00084-f003:**
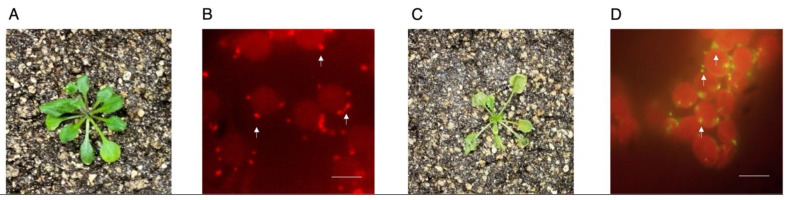
Leaf mesophyll mitochondria depolarize upon heat shock treatment. (**A**) Leaf mesophylls from plants grown under normal temperature conditions were stained with JC-1 and imaged. (**B**) Mitochondria appear as high-intensity fluorescent red particles surrounding chloroplasts. (**C**) Leaf mesophylls from plants grown under normal temperature conditions and treated at 37 °C for three hours were stained with JC-1 and imaged. (**D**) Mitochondria appear as high-intensity green particles surrounding chloroplasts. Arrows mark selected mitochondria. Scale bar: 10 µm. Representative images *n* = 5.

## Data Availability

Data reported in this study are available upon institutional material transfer agreement approval upon request by academic researchers for non-commercial reasons.

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
