# Peer review of "Leaf Mesophyll Mitochondrial Polarization Assessment in Arabidopsis thaliana"

_mps, 2021, doi:10.3390/mps4040084_

Round 1
Reviewer 1 Report
Dear Authors,
I have an honor to review manuscript submitted to the Methods and Protocols in MDPI, entitled “Leaf mesophyll mitochondrial polarization assessment in Arabidopsis thaliana”.
- I see and believe in the potential of presented method -a fast and straightforward protocol to assess changes in leaf mitochondrial membrane potential. I would like to underline the analysis in situ ;
Unfortunately, I have my doubts about novelty aspect of presented methodology. Authors did almost not described the prospects coming from these methods; Please, describe more the application potential of this method;
How about fluorescence signal quantification coming from “improved” mitochondria condition ?
- I suggest to improve the procedure and material details: please detailed the recipe as much as it will be possible to make possible to repeat your procedure.
Moreover, Please, expand abbreviations like JC-1 carbocyanine dye, and so on… , precise term “50% bleach”, or what about “stain” and “wash buffer” (point 5).
I see also one big disadvantage of presenting procedure , for 45min leaf saining without epidermis and 10 minutes for mounting, and then we analyse the effect – I hazard to suggest that we do not receive the potential state with stress elimination !
I also warmly suggest to improve the microscope micrographs quality
- Authors concentrated on thaliana leaves, what about different leaf types-it means thicker with modified tissue layers ? Is it possible some kind of modification adjusting to other species like Nicotiana, or representant of Fabaceae ?
Author Response
I see and believe in the potential of presented method -a fast and straightforward protocol to assess changes in leaf mitochondrial membrane potential. I would like to underline the analysis in situ ;
Answer: Thank you very much for your comments.
Unfortunately, I have my doubts about novelty aspect of presented methodology. Authors did almost not described the prospects coming from these methods; Please, describe more the application potential of this method;
Answer: Previous reports have assessed mitochondria from tissues with low mature chloroplast yield including shoot apical meristem, leaf primordium as well as stems and root tips (for example PMID: 18799659). Nevertheless, this is the first report -to our knowledge- to show that mitochondria from leaf mesophyll cells can be easily stained in living tissue without the need of complex manipulations. The manuscript now includes potential applications of the proposed methodology on line 18 as follows: “This method can be applied to determine mitochondrial status and morphology in plants undergoing the first stages of growth and under conditions where other mitochondrial assessment methods are difficult to implement.”
How about fluorescence signal quantification coming from “improved” mitochondria condition ?
Answer: We have not assessed mitochondrial signal under conditions thought to improve membrane polarization. The reason we did not do so is because our control samples have already maximal red fluorescence indicating maximal mitochondrial transmembrane potential. However, we did perform experiments from mitochondria under deteriorated conditions (i.e. heat shock and in the presence of an uncoupler). Under these conditions, mitochondria appear as green fragmented particles surrounding chloroplasts (Fig. 2).
I suggest to improve the procedure and material details: please detailed the recipe as much as it will be possible to make possible to repeat your procedure.
Answer: The manuscript now includes further details in the procedure, materials and buffer sections.
Moreover, Please, expand abbreviations like JC-1 carbocyanine dye, and so on… , precise term “50% bleach”, or what about “stain” and “wash buffer” (point 5).
Answer: Thank you for noticing this. The manuscript now includes definitions for used abbreviations and precise terms for reagents.
I see also one big disadvantage of presenting procedure , for 45min leaf saining without epidermis and 10 minutes for mounting, and then we analyse the effect – I hazard to suggest that we do not receive the potential state with stress elimination !
Answer: We agree with this concern. However, our control samples do show a high mitochondrial membrane potential consistent with functional organelles. In addition, current methods including staining thin sections or protoplasts imply even greater stress conditions for the cells and even longer procedures (up to hours).
I also warmly suggest to improve the microscope micrographs quality
Answer: The revised version of the manuscript includes new images with the highest resolution achievable with our microscope.
Authors concentrated on thaliana leaves, what about different leaf types-it means thicker with modified tissue layers ? Is it possible some kind of modification adjusting to other species like Nicotiana, or representant of Fabaceae ?
Answer: We have also assessed mitochondrial polarization in leaf mesophylls from 3 week-old Nicotiana tabacum with promising results. However, we believe this method must be optimized in a species-specific manner and currently our method provides excellent results with Arabidopsis thaliana.
Reviewer 2 Report
The manuscript by Flores-Herrera1et al.; focused on techniques to visualize plant mitochondria dynamics during various stresses. Research on plant mitochondria is limited; thus, new technics are very welcome. Generally, I find the manuscript interesting and well-written. The protocol could be very usefull to other scientists in the field.
Author Response
Thank you for your comments.
Reviewer 3 Report
In this manuscript, the authors check the leaf mesophyll mitochondrial polarization assessment in Arabidopsis thaliana.
In the current methods for leaf, mitochondrial membrane potential determinations have been traditionally performed in thin mesophyll sections, isolated protoplasts, or fluorescent protein-expressing transgenic plants. However, this may limit the amount of information obtained about overall mitochondrial morphology in intact leaves. Here we detail a fast and straightforward protocol to assess leaf mitochondrial membrane potential changes associated with mitochondrial dysfunction in the model plant Arabidopsis thaliana. This protocol permits mitochondrial shape, dynamics, and polarity assessment in leaves subjected to diverse stress conditions. The manuscript is written very well, and this protocol has merit to be published in this journal in the current form.
Change at L12 from mesophyll thin sections to thin mesophyll sections.
Author Response
Change at L12 from mesophyll thin sections to thin mesophyll sections.
Answer: Thanks for noticing, we corrected this mistake.
Round 2
Reviewer 1 Report
Authors significantly improved manuscript according suggestions and provide satisfactoctory answers